# New Predictive Biomarkers for Ovarian Cancer

**DOI:** 10.3390/diagnostics11030465

**Published:** 2021-03-07

**Authors:** Ghofraan Abdulsalam Atallah, Nor Haslinda Abd. Aziz, Chew Kah Teik, Mohamad Nasir Shafiee, Nirmala Chandralega Kampan

**Affiliations:** Department of Obstetrics and Gynaecology, Universiti Kebangsaan Malaysia Medical Centre, Kuala Lumpur 56000, Malaysia; ghofraan.a@gmail.com (G.A.A.); norhaslinda.abdaziz@ppukm.ukm.edu.my (N.H.A.A.); drchewkt@gmail.com (C.K.T.); nasirshafiee@hotmail.com (M.N.S.)

**Keywords:** ovarian cancer, biomarkers, tumour mutation burden, DNA repair pathways, cell-cycle-related genes

## Abstract

Ovarian cancer is the eighth-most common cause of death among women worldwide. In the absence of distinctive symptoms in the early stages, the majority of women are diagnosed in advanced stages of the disease. Surgical debulking and systemic adjuvant chemotherapy remain the mainstays of treatment, with the development of chemoresistance in up to 75% of patients with subsequent poor treatment response and reduced survival. Therefore, there is a critical need to revisit existing, and identify potential biomarkers that could lead to the development of novel and more effective predictors for ovarian cancer diagnosis and prognosis. The capacity of these biomarkers to predict the existence, stages, and associated therapeutic efficacy of ovarian cancer would enable improvements in the early diagnosis and survival of ovarian cancer patients. This review not only highlights current evidence-based ovarian-cancer-specific prognostic and diagnostic biomarkers but also provides an update on various technologies and methods currently used to identify novel biomarkers of ovarian cancer.

## 1. Introduction

Ovarian cancer is the eighth-most commonly occurring cancer in women worldwide, contributing to 300,000 new cases globally in 2018. Ovarian cancer is often asymptomatic in the early stages; therefore, it is considered a silent killer. The signs and symptoms become more apparent as the cancer advances. For this reason, ovarian cancers have the highest mortality rate among the gynaecological diseases. Therapeutic intervention for a cure is possible in the early stages of the disease, and this explains the direct relation between early diagnosis and survival [1,2]. In contrast to a five-year survival rate of 95% at an earlier stage, the high case fatality ratio illustrates the reality that most cases of ovarian cancer are diagnosed in an advanced stage of the disease (stage III or IV), with a five-year survival rate of less than 30% [3]. To date, no screening approach has been recommended for early stage identification. It will also be important to establish tumour markers that could be used to diagnose cancer in the early stages in order to improve the survival of these women.

Epithelial ovarian cancer (EOC) is the commonest type, highly heterogeneous, and often linked to genetic instability [4]. Serous-type ovarian cancer comprises up to 70% of EOCs. The World Health Organization (WHO) classification system, published in 2014, reclassified serous ovarian cancer into two separate entities: high-grade and low-grade [5]. Low-grade serous ovarian cancer (LGSOC) represents 5–10% of serous ovarian cancers. LGSOC is well-differentiated and can be identified in the early stages with the overexpression of serous borderline precursor lesions, such as BRAF and KRAS mutation, and the near absence of the tumour protein p53 mutation (TP53) [6]. High-grade serous ovarian cancer (HGSOC), meanwhile, is the most common and aggressive form of EOC subtype and accounts for the majority of ovarian-cancer-related deaths [7]. Unlike LGSOC, HGSOC has no known precursor lesion in the ovary, with many studies suggesting its development from intraepithelial carcinomas in the fallopian tube [8]. Approximately 95% and 50% of HGSOC cases have TP53 mutations and homologous recombination (HR) deficiency, respectively [5,9]. Clinically, women with HGSOC tend to be older and have poor survival, while those with LGSOC are younger and tend to survive longer [10]. Although HGSOC patients have initial sensitivity to first-line chemotherapy, almost all patients develop chemoresistance. LGSOC is a slow-growing tumour and generally resistant to chemotherapy but responds well to aggressive debulking surgery. LGSOC has shown improved survival rates when treated with alternative treatments such as angiogenesis inhibitor and MEK inhibitor therapy [11]. The heterogeneity of ovarian cancer is a major obstacle in discovering novel biomarkers to aid early detection. As new therapies such as Poly ADP Ribose Polymerase (PARP) inhibitors have improved survival in women with HGSOC, specifically in those with the BRCA mutation, the discovery of molecular biomarkers is becoming important. As ovarian cancer targets various genetic and putative molecular cell signalling pathways, understanding the related proteins and genes will provide an opportunity to unravel new biomarkers. In this review, we summarise recently discovered potential biomarkers to improve the diagnosis of, and prognosis for ovarian cancer, and revisit the clinically available biomarkers and their precision of use in the management of EOC. In addition, this article is designed to play a role as a platform to search for the most effective laboratory technology and methods available to discover novel ovarian biomarkers.

### Search Strategy of Review

The keywords and terms of the major concepts for this review included ovarian cancer, ovarian signalling pathway, ovarian cancer diagnosis and prognosis, and current and/or new ovarian cancer biomarkers, which were developed and combined to form the search strategy. In Section 2.1, Ovarian Cancer Cell Signalling Pathways and Their Clinical Utility, we included more specific keywords of ovarian cancer molecular pathways (e.g., tumour mutation burden, DNA repair pathways, and cell-cycle-related genes). A systematic search of Google Scholar, PubMed, Web of Science, and Sci-Hub databases was combined for relevant publications, from the time of their inception to December 2020. Results were merged using reference management software (Endnote, Thomson Reuters). The findings of relevant studies are summarised in Table 1 and Table 2.

## 2. An Overview of Ovarian Cancer Biomarkers and Cell Signalling Pathways

Numerous cancer genes and protein have been studied as diagnostic, prognostic and targeted therapy biomarkers in ovarian cancer management. Some of the most common biomarkers and their clinical utility are summarised in Table 1, while their sensitivity and specificity at early stages (stage I–II) of the disease are summarised in Figure 1. These biomarkers are already known to be overexpressed or deregulated in various signalling pathways in women with ovarian cancer [52]. Evidently, ovarian cancer is not a common disorder but includes a heterogeneous community of tumours that affect various molecular genetic mechanisms and putative molecular signalling pathways [53]. The main signalling pathways targeted for ovarian cancer diagnosis, progression and treatment are outlined in the following sections.

### 2.1. Ovarian Cancer Cell Signalling Pathways and Their Clinical Utility

#### 2.1.1. BRCA1 and BRCA2 Pathway

In over 15% of women with HGSOC are associated with breast cancer 1 (*BRCA1*) and *BRCA2* oncogenes, inherited as germline mutations. The *BRCA1* and *BRCA2* genes are located on chromosomes 17q21 and 13q12, respectively [54,55]. Wild-type *BRCA1/2* genes are critical for DNA repair by the homologous recombination (HR) pathway; hence their deletion causes genomic instability and predisposes affected women to familial breast and ovarian cancers [56,57]. In addition, ovarian cancers with defective *BRCA1/2* genes are especially susceptible to agents that induce DNA double-strand breaks (DSBs) [58] and DNA interstrand cross-links, such as platinum compounds (e.g., cisplatin and carboplatin) and polymerase (PARP) inhibitors (e.g., Olaparib, Iniparib and Veliparib) [39,59]. It is therefore conceivable that secondary or reversion mutations of the *BRCA1/2* genes by multiple complex mechanisms can benefit the DNA repair of HR and increase the survival of the tumour cells, thus triggering resistance to these compounds [60]. As demonstrated in a SOLO-1 clinical trial, the use of PARP inhibitors as maintenance therapy represents a paradigm shift in ovarian cancer management, specifically in HGSOC women with *BRCA* mutations. There is a 70% lower risk of disease progression and a longer time before subsequent chemotherapy with PARP inhibitor maintenance therapy for 2 years [61].

#### 2.1.2. MAPK/ERK Pathway

In ovarian cancer, the downstream proteins of the MAPK/ERK pathway play a pertinent role in promoting the migration and invasion of the disease and may therefore contribute to metastasis and chemoresistance. Genetic mutations such as BRAF and KRAS frequently contribute to the activation of the MAPK/ERK pathway, which is usually observed in LGSOC but rarely in HGSOC [62]. The mitogen-activated protein kinase (MAPK) pathway consists of three sequentially activated protein kinases responsible for the regulation of cell proliferation, cell differentiation, and cell death in humans. In response to external stimuli, including hormones such as follicle-stimulating hormone (FSH) and luteinising hormone (LH), growth factors [63], cytokines and chemotherapeutic chemicals, the MAPK pathway is activated by interactions with a small GTPase and/or phosphorylation by protein kinases downstream from cell surface receptors, such as G-protein coupled (Gonadotropins and Gonadotropin-releasing hormones) and tyrosine kinase (RTK) receptors [63]. MAPKs often play a role in promoting ovarian cancer cell growth through membrane receptor signals for gonadotropins-FSH and LH, frequently expressed in ovarian carcinoma cells, and may therefore lead to signal transduction through MAPK [64].

Clinical trials with targeted BRAF and MEK inhibitors, such as selumetinib, have been shown to achieve long-term progression free survival (PFS) and a complete response in advanced LGSOC [65]. In Gynecologic Oncology Group study 239, a phase II trial of selumetinib showed a 15% overall response rate and a median PFS of 11 months achieved by previously treated LGSOC patients (*n* = 52). This a modest improvement compared to PFS of seven months with traditional chemotherapy [66].

#### 2.1.3. EGFR/AKT Signalling Pathway

The epidermal growth factor receptor (EGFR) is expressed in 70% of ovarian cancers [67]. Various ligands, such as EGF and TGF, may be activated and play a part in promoting and inhibiting tumour survival [68,69]. EGFR is also involved in tumour infiltration, metastasis and angiogenesis [70]. AKT is a major downstream signalling element for EGFR. Upon binding to EGFR, AKT is activated by phosphorylation. AKT is regularly overexpressed in ovarian cancer and is associated with aggressive tumour activity and poor prognosis. As the EGFR/AKT pathway is implicated in different facets of cancer proliferation, such as angiogenesis and metastases, it is currently seen as an attractive option for therapeutic intervention. Cetuximab (Erbitux) was the first anti-EGFR monoclonal antibody tested in several solid tumours, such as breast, colorectal, head and neck, renal, gastrointestinal stromal tumours (GISTs) and lung cancers. Although there is evidence of the therapeutic efficacy of anti-EFGR in these solid tumours, treatment with anti-EFGR agents has induced only low treatment response in ovarian cancers. In future, more comprehensive research needs to be done to unravel the protein and gene molecules along the complex EGFR signalling pathway in ovarian cancer to identify biomarkers that can accurately gauge the sensitivity of EGFR-targeted therapeutic agents.

#### 2.1.4. Integrin Inhibitor Pathway

Integrins are heterodimeric adhesion receptors expressed on the cell surface, and consist of two noncovalently bound subunits, namely an α subunit and a β subunit [71]. The latest research has examined the use of integrin inhibitors as possible therapeutic agents in ovarian carcinoma. Several preclinical studies on various integrin antagonists have shown their effectiveness in blocking tumour progression. Integrin antagonists inhibit tumour progression by affecting both tumour cells and tumour-associated host cells, especially the angiogenic endothelium. The initial step in ovarian carcinoma dissemination occurs with the attachment of cancer cells onto the peritoneal surface via integrins, so targeting integrins seems a rational therapy approach [16].

Although no integrin inhibitors have yet shown desirable efficacy results, integrin-targeted therapies continue to be a promising pathway for further clinical investigation. In cancer progression, as more than one integrin pathway is usually involved, it was not surprising to see that agents targeting only a single integrin, such as αvβ3 and α5β1, failed to show clinical benefits in metastatic cancer treatment. Therefore, a combination of different integrin receptor pathways is likely to be show therapeutic efficacy in clinical trials and should be further explored [16].

#### 2.1.5. GRP78 Expression Pathway

A recent investigation has suggested GRP78 as a drug delivery system for ovarian cancer cells. GRP78 upregulation is a cellular reaction process, caused by endoplasmic reticulum stress, and usually seen in tumour cells. Since GRP78 is abundantly present on ovarian cancer cell surfaces, recent research indicates the use of GRP78 as a delivery mechanism for cytotoxic substances [17].

#### 2.1.6. P38 Alpha Pathway

The p38alpha pathway has recently been the focus of cancer research. Small compound inhibitors of p38alpha have already been evaluated in clinical trials, showing that the pharmacological blockade of p38 reduced the growth and viability of ovarian cancer cells [18]. The pharmacological blockade of p38 in ovarian cancer has been shown to induce the formation of large autophagic vacuoles containing cytoplasmic glycoproteic material and mitochondrial debris, suggesting that the inhibition of p38 caused autophagic cell death in ovarian cancer cells. This biomarker should be further investigated as a future therapeutic opportunity for ovarian cancer [72].

### 2.2. Current Biomarkers Associated with Diagnosis, Progression and Treatment Response of Ovarian Cancer

#### 2.2.1. Carbohydrate Antigen 125 (CA125)

Cancer antigen 125 or carcinoma antigen 125, also known as MUC16, is a protein encoded by the MUC16 gene [73]. Clinically, it is used as a diagnostic test to measure the amount of the protein CA125 in the serum. In most laboratories, the normal value for CA125 is 0 to 35 units/mL.

Up to 80% of women diagnosed with late-stage epithelial ovarian cancer have elevated CA125 levels in their serum [22]. Unfortunately, CA125 has limited usefulness in detecting ovarian cancer in the early stages, as only 50% of these cases had elevated CA125 levels [23]. In addition, many other conditions can also cause the elevation of CA125 levels, including endometriosis, liver cirrhosis, normal menstruation, pelvic inflammatory disease and uterine fibroid. Therefore, this antigen lacks the specificity and sensitivity to be considered a reliable biomarker for the early detection of ovarian cancer [43]. CA125 is found to be more sensitive and specific in postmenopausal women than in premenopausal women. Serum CA125 is incorporated into the Risk Malignancy Index (RMI) algorithm. The RMI is widely used as a risk assessment of ovarian malignancy in clinical practice. The RMI score is generated by the simplified serum CA125 level regression equation, the menopausal status score and the ultrasound features score (RMI = ultrasound findings × menopause status × CA125 U/mL). The use of the RMI has a higher sensitivity of 87% and a specificity of 97% for ovarian cancer detection compared to CA125 alone [74].

In a recent study using a training and confirmation cohort, four existing clinical tests available for the diagnosis of ovarian cancer (RMI score and ROMA, CA125 and HE4) and a panel of 28 immunosoluble biomarkers from 66 patients undergoing surgery for suspected ovarian cancer were assessed through a multiplex immunoassay. Using a two-step triage model for women with presumed ovarian mass, IL-6 > 3.75 pg/mL was established as the main triage, supplemented by standard testing (CA125 or RMI score) for ovarian cancer in patients greater than CA125 or RMI alone (misclassification rate 4.54–3.03 percent vs. 9.09–10.60). Therefore, in conjunction with traditional studies, IL-6 can be a beneficial therapeutic biomarker for the triage of patients with potential malignant ovarian mass [26]. The reproducibility of IL6 measurement may be a challenge in clinical practice as IL6 level can increase if infections and or inflammatory conditions occur.

#### 2.2.2. Osteopontin (OPN)

Osteopontin is an adhesive glycophosphoprotein secreted by activated T lymphocytes, macrophages, and leukocytes, and found in the extracellular matrix, sites of inflammation and body fluids [75] Osteopontin is not only expressed in ovarian cancer but also in endometrial, cervical, breast, colorectal, nonsmall cell lung, prostate, hepatocellular and gastric cancer. OPN is associated with tumour progression, invasion and metastasis. In 2001, OPN was identified with a cDNA microarray system using RNA isolated from several ovarian cancer cell lines, with surface epithelial cells as controls [50,76]. The levels of OPN were also significantly higher (*p* < 0.001) in the plasma samples of 51 women with EOC (486.5 ng/mL, *n* = 51) compared with the healthy controls (147.1 ng/mL, *n* = 107), benign ovarian disease (254.4 ng/mL, *n* = 46), and other gynaecological cancers (260.9 ng/mL, *n* = 47) [77].

In addition, OPN has been utilised to predict the progression of disease in advanced EOC, as the prognosis of patients with peritoneal spread is poor. The levels of osteopontin in 32 out of 40 peritoneal metastatic biopsies were found to be significantly elevated compared to the levels found in primary ovarian tumour tissues among women with Stage III EOC [78]. In addition, the elevated OPN levels were independently correlated with extremely poor prognosis among these women (*n* = 32), whereas 75% of the women found with no increase in OPN levels had a 36-month survival rate (*n* = 8). Furthermore, the high levels of osteopontin could be measured in the urine samples of patients with high-grade ovarian cancer, so this test could potentially be used clinically as a noninvasive tool for the early diagnosis of ovarian cancer [79].

#### 2.2.3. Kallikreins (KLKs)

Kallikreins are a subgroup of serine proteases with different physiological roles. The human kallikrein gene family has now been entirely defined to include 15 members on chromosome 19q. They are expressed in epithelial and endocrine tissues regulated by hormones in cancer and they are shed and detected in human body fluids [34]. Therefore, many studies have been carried out to find their role in cancer diagnosis and prognosis. A total of 12 out 15 KLKs are upregulated in ovarian cancer, with some KLKs correlating to poor prognosis and late-stage disease (4–7, 10 and 15), as well as chemoresistance (KLK 4 and 7) to a first line paclitaxel agent [80].

In a study by Luo et al., the preoperative serum level of human kallikrein (hk10) in 146 patients with ovarian cancer was significantly elevated compared to 97 healthy women and 141 women with benign gynaecological diseases [81].

#### 2.2.4. Bikunin

Bikunin is a multifunctional glycoprotein, which mediates the suppression of tumour cell invasion and metastasis. The measurement of bikunin levels in the tissue of patients with malignant diseases has been introduced as a simple diagnostic tool for the evaluation of the prognosis. High preoperative bikunin levels have been reported to be a strong favourable prognostic marker for ovarian cancer [13]. Matsuzaki et al. found, in an extensive study, that bikunin protein in the plasma of women with ovarian cancer (*N* = 327), compared to those with benign ovarian mass (*N* = 200) and healthy controls (*N* = 200), may be useful in evaluating the prognosis of the disease.

A low bikunin level (≤11.5 ug/mL) was found to be associated with the late-stage (Stage III/IV) disease, the presence of large residual tumours (>2 cm) and poor response to chemotherapy. The median survival time was also shorter, at 26 months compared to 60 months, in those with high levels of bikunin (*p* = 0.002), thus corresponding to a 2.2-fold higher risk of dying (hazard ratio, 0.45; *p* = 0.023) [13]. Measuring levels of bikunin in plasma is easy and relatively inexpensive; therefore, it has the potential to be included as a prognostic biomarker for ovarian cancer. However, there is a significant overlap in bikunin levels across cancer, benign and healthy controls, which needs to be further investigated before it can be of clinical use.

#### 2.2.5. Human Epididymis Protein 4 (HE4)

Also known as WAP 4-disulphide core domain 2 (WFDC2), HE4 was first introduced as an ovarian cancer biomarker in 1999 [36]. The expression of HE4 is associated with cancer cell adhesion, migration and tumour growth, which can be related to its effects on the EGFR-MAPK signalling pathway [82]. Many studies suggested that HE4 is absent in normal ovarian surface epithelium but is expressed specifically in 100% of human endometrioid epithelial ovarian cancers (*n* = 16) and 93% of serous ovarian carcinomas stained for HE4 (*n* = 60) [57]. An ELISA analysis of serum HE4 levels in 37 patients with ovarian cancer, compared with 65 healthy controls, showed that HE4 had the same specificity and sensitivity as CA125 and detected fewer false positives in patients without a malignant disease [74].

The marker HE4 is significantly increased in ovarian and endometrial cancer, but not in endometriosis. HE4 can be increased, although it is less frequently elevated than CA125 in patients with benign disease, especially in premenopausal patients. The alternate probability of a malignancy algorithm (ROMA) blends the values of CA125 and HE4 with menopausal status in the predictive index and has been shown to stratify patients into high and low risk categories, with differing outcomes across many trials [83]. 

#### 2.2.6. Vascular Endothelial Growth Factor (VEGF)

VEGF is a vascular permeability factor that is a key regulator of physiological and pathological angiogenesis, and makes a major contribution to tumorigenesis [84]. VEGF levels are known to be elevated in patients with ovarian cancer and contribute to the accumulation of ascites [85]. An analysis associated with VEGF levels in the preoperative sera of 314 patients with ovarian cancer recorded that higher VEGF levels were separately correlated with shorter survival periods [86]. In addition, tumour samples from 18 patients with advanced stage serous epithelial ovarian cancer were evaluated for VEGF expression by a reverse-transcriptase polymerase chain reaction (RT-PCR) [87]. It was demonstrated that 12 samples were found to be strongly positive, whereas six samples had low/negative VEGF expression. The median survival was longer, at 60 months in the VEGF-low/negative group compared to 28 months in the VEGF-positive group (*p* = 0.058).

Bevacizumab, the first and most studied anti-VEGF agent, when used as maintenance therapy following surgical debulking and first-line chemotherapy, led to significant improvements in the progression-free survival of patients with ovarian cancer but did not have an impact on the survival. Bevacizumab, in addition to PARP inhibitors, is currently being studied in Phase III PAOLO-1/ ENGOT-ov25 trials and has shown promising results, with reduced risk of disease progression by 41% overall, and by 69% in the subset of women with *BRCA*-mutated disease [88].

#### 2.2.7. Human Prostasin (PSN)

PSN is a trypsin-like proteinase (40 KDa) found on chromosome 16p11.2. It plays a major role in the activation of epithelial sodium channels and in the reduction of invasive prostate and breast cancers in vitro [89]. Similarly, the epidermal tight junction forming and terminal differentiation are related to the matriptase-prostasin proteolytic pathway [90].

The potential use of prostasin as a novel biomarker for ovarian carcinoma was proposed by Mok et al. using microarray technologies to classify upregulated genes for secretive proteins [91]. The findings revealed an overexpression of PSN in malignant epithelial ovarian cells and stroma, relative to standard ovary tissue, with a sensitivity and specificity of 51.4% and 94%, respectively [92]. Gene expression analysis indicated that PSN was expressed in ovarian cancer at levels more than 100 times greater than those found in normal or benign ovarian lesions. This overexpression signature was found in the early stages of ovarian cancer and maintained in the higher stages and grades [93]. Costa et al., on the other hand, reported a slightly higher overexpression of mRNA prostasin in freshly frozen ovarian cancer tissues than in usual controls. Thus, it has the ability to be used clinically as a differential diagnostic marker for ovarian cancer [93]. In another study by Mok et al., the combination of CA125 and prostasin gave a sensitivity of 92% and a specificity of 94% for detecting ovarian cancer [91].

#### 2.2.8. Creatine Kinase B (CKB)

Creatine kinase plays a crucial function in the energy homeostasis of vertebral cells. CKB is a cytosolic isoform of creatine kinase that displays upregulated expression in a number of cancers. It has been reported that certain ovarian cancer tissues have improved protein CKB expression [94]. In addition, CKB decreased the intake of glucose and lactate, and improved the ROS output and consumption of oxygen. As a result, it was indicated that the suppression of CKB induced G2 arrest in the cell cycle through the PI3K/AKT and AMPK pathways. Clinically, this mechanism has helped clinicians to use this biomarker in cancer cell survival and tumour progression. CKB activity measured in preoperative serum samples was higher in women with ovarian cancer (*N* = 45), compared to those with benign ovarian mass (9.6 U/L, *N* = 49) and healthy controls (8.5 U/L, *N* = 37), *p* = 0.0096 [27]. CKB is highly expressed in early stage ovarian tumour tissues and is, therefore, a potential biomarker for the early detection of ovarian cancer; it should be further investigated [27].

#### 2.2.9. Mesothelin

Mesothelin, a tumour differentiation antigen found in mesothelial pleura, peritoneum and pericardium, was discovered in 1996 at the National Cancer Institute [95]. Mesothelin is widely expressed in many tumours, including 70% of ovarian cancers. Several mesothelin-directed treatments have been studied in clinical trials, including antimesothelin immunotoxins and antibody-drug conjugates (ADC) [96]. Quanz et al. showed the activity of anetumabravtansine in conjunction with conventional chemotherapy in ovarian cancer models. Anetumabravtansine is an ADC that produces a human antimesothelin antibody conjugated by a reducible disulphide linker to the DM4maytansinoid tubulin inhibitor. Both in vitro and in vivo experiments have indicated the selective activity of anetumabravtansine in injecting new expression cells and tumours, including low-sensitivity (68.2%) and high-specificity (80.5%) ovarian cancer [97]. In animal models with ovarian cancer, treatment with anetumabravtansine exhibits improved potency in combination with carboplatin, compared to either drug alone [97]. Similarly, Anetumabravtansine also demonstrates enhanced antitumour efficacy when combined with Bevacizumab, an anti-VEGF agent. A phase 1b study (NCT02751918) using anetumabravtansine in combination with pegylated liposomal doxorubicin in ovarian cancer patients is ongoing.

#### 2.2.10. Apolipoprotein A-I (apoA-I)

ApoA-I is a high-density lipoprotein (HDL) and apolipoprotein A-I in plasma. Apo A-I levels have been reported to decrease in the sera of patients with ovarian cancer [44]. A multiplexed magnetic nanoparticle-antibody conjugates (MNPs-Abs) based fluorescence spectroscopic system analysis combining CA125, β2-M and ApoA1 for the early detection of ovarian cancer performed by Pal et al. found that while CA125 detection only identifies 50–60% of early stage ovarian cancer, the combination of the three biomarkers achieved high sensitivity (94%) and high specificity (98%) in distinguishing early stage ovarian cancer patients from healthy individuals [98]. This proposed multiplexed panel assay is also cost-effective, and further clinical investigation should be conducted to develop a clinically beneficial test kit.

#### 2.2.11. Transthyretin (TTR)

TTR is a natural serum protein synthesised mostly in the liver [99]. It attaches and transports the thyroid hormones and retinol protein binding to the retinal complex [100]. Low TTR serum levels were found in ovarian cancer and used with other biomarkers to detect ovarian cancer [29,101]. Using liquid chromatography with tandem mass spectrometry, Kozak et al. found that TTR, in combination with beta-haemoglobin, apolipoprotein AI, transferrin and CA125, significantly improved the detection of early stage ovarian cancer [101]. TTR was found to be an important marker for the detection of stage I–II ovarian cancer, with a sensitivity and specificity of 78.6% and 68.8%, respectively.

#### 2.2.12. Transferrin

Transferrin is essentially synthesised in hepatocytes and responsible for delivering plasma iron to the cell. It plays a major role in cell division and proliferation [43]. Ahmed et al. documented the downregulation of transferrin in the sera of patients with ovarian cancer [102]. In another case-control study, the level of transferrin was measured using an immunological turbidimetric assay in the sera of 37 women with ovarian cancer and compared to those with benign ovarian diseases (*N* = 31) and age–matched healthy controls (*N* = 31). It was found that the use of the biomarker transferrin as a detection tool for ovarian cancer has only low sensitivity and specificity, at 72.9% and 74.1%, respectively [40]. Therefore, transferrin needs to be used in combination with other biomarkers to achieve clinical significance.

### 2.3. Highlighting the Most Common Biomarker Combinations for the Management of Ovarian Cancer

It has been shown that a combination of certain potential biomarkers could significantly improve the detection and management of ovarian cancer. One of the most common combinations is the use of CA125 and PSN together, which resulted in an improved sensitivity (92%) and specificity (94%), compared with CA125 alone (sensitivity of 64.9% at a specificity of 94%) and PSN (sensitivity of 51.4% at a specificity of 94%) [92]. In addition, a combination of Apo-A1, TT, Connective tissue activating peptide III (CTAPIII) and CA125 achieved a sensitivity of up to 84% and a specificity of 98% in distinguishing women with early stage ovarian cancer from healthy individuals [40]. When ApoA1 was combined with CA125 and TTR, not only was a significant improvement observed in the overall sensitivity and specificity, but the panel was also sufficient for maximum discrimination between noncancer, stage I–II and all stages (I–IV) of ovarian cancer [103].

Similarly, Kozak et al. revealed that the TTR, Hb, ApoAI and TF biomarkers, when integrated with CA125, should significantly improve the detection of early stage ovarian cancer [104]. In addition, Kim et al. suggested the benefit of the combination of TTR, apolipoprotein A1 and CA125 in the diagnosis of ovarian cancer. The combination of CA125, transferrin, TTR and ApoA1, using a proteomic analysis, yielded a sensitivity of 89% at a specificity of 92% for the early detection of ovarian cancer [105].

## 3. Emerging Predictive Ovarian Cancer Biomarkers

### 3.1. An Overview of the Molecular Approaches to the Discovery of New Ovarian Cancer Biomarkers

To identify new biomarkers requires intensive, specific technologies and methods for the detection of molecules, genes and proteins in human body fluids and tissues. This section addresses some of the most common technologies recently used to discover new biomarkers of ovarian cancer.

#### 3.1.1. Whole Genome Analysis

Comparative genomic hybridisation (CGH) is a full genome assay that detects gene copy gains or losses. This assay found multiple chromosome regions with an abnormal gene copy number for ovarian cancer [106]. Genomic expression profiling studies on epithelial ovarian cancer of different histology have elucidated not only global gene expression profiles and signalling pathways to distinguish and characterise each subtype of cancer, but also potential prognostic indicators [103].

#### 3.1.2. Transcription Profiling

Transcription profiling, also known as ‘expression profiling’, is one of the most common types of analysis. It includes the quantification of the gene expression of several genes in the transcription (RNA) of cells or tissue samples. Quantification can be achieved by collecting biological samples and extracting RNA after a treatment or at fixed time-points in a time-series, thus generating “snapshots” of the expression patterns. Various histological subtypes of ovarian cancer are associated with different prognoses, and several transcription profiling studies have focused on the discovery of markers that can differentiate between subtypes. A number of transcription profiling studies have shown that a certain correlation between the subtypes of gene expression signature still exist, indicating some of the shared mechanisms underlying ovarian carcinogenesis [107]. Transcription profiling studies have established markers that could predict patient survival [108].

#### 3.1.3. MicroRNA Profiling

MicroRNA was first observed in *Caenorhabditis elegans* in 1993 [108]. These RNAs contain 19 to 24 nucleotides and do not encode proteins. They interact with the three untranslated target mRNAs’ region, leading them to target mRNA degradation and inhibition [109].

MicroRNAs have been shown to be differentially expressed in tumours versus normal tissues in a number of solid and hematopoietic tumours. In certain cases, distinct microRNA signatures can reliably differentiate tumours from normal tissues and are associated with disease outcomes. In addition, a study examining microRNA signatures in a variety of tumour types indicated that the expression pattern of a relatively small number of microRNAs (approximately 200) was more reliable than cDNA arrays in the classification of human cancers [109]. These studies strongly suggest that microRNA profiling may have significant potential for cancer diagnosis and prognosis.

#### 3.1.4. Proteomic Profiling

One major drawback of transcription profiling studies is that the changes in the mRNA level do not always result in changes in protein levels. Proteomic profiling has also recently been recognised as the most direct approach in the search for diagnostic and prognostic ovarian cancer biomarkers, hence mass spectrometry is one of the key methods used in proteomic profiling. The proteomics of ovarian cancer can be done using two techniques. One is the detection of distinct proteomic peptide patterns in cancer samples [1]. The second is to classify individual peptides that can distinguish between cancer and normal samples.

In addition to the proteomic profiling of serum and plasma samples, profiling can also be performed on other body fluids. These include glycosylated eosinophilic-derived neurotoxins and COOH-terminal osteopontin fragments, and the most established is ovarian cancer ascites proteome, where approximately 80 biomarkers have been discovered and identified as potential markers for early stage ovarian cancer detection [110].

### 3.2. Emerging Biomarkers Associated with Ovarian Cancer Diagnosis and Prognosis

The diagnosis of ovarian cancer is currently focused on restricted imaging techniques and the concentration of certain biomarkers circulating with established levels of sensitivity and specificity. New biomarkers are required to complement and improve the efficacy of the existing clinical tests. New biomarkers, including circulating DNA tumours, serum tumour proteins, circulating cancer cells or serum metals such as Cu and zinc, are emerging to complement the clinically available diagnostic methods [19,111]. A summary of these new biomarkers is shown in Table 2.

#### 3.2.1. Cu Isotope

The concentration of Copper (Cu) in the bloodstream is regulated by two major organs: it is absorbed by the intestine and transported to the liver [112]. Changes in the concentration of Cu influenced by modified metabolic processes can affect health and disease [46]. In recent study looking at copper composition ^65^Cu/^63^Cu ratios (Δ^65^Cu), in blood samples from 44 ovarian cancer patients, and 13 ovarian biopsies using multicollector inductively coupled plasma mass spectrometry, a connection has been demonstrated to cancer progression [113]. The copper isotope ratio δ^65^Cu in the plasma of ovarian cancer patients (*n* = 44) was shown to be lower, in comparison to the levels among healthy donors (*n* = 48), suggesting that the serum was enriched with ^63^Cu.

#### 3.2.2. Exosomes

Exosomes are endocytic and heterogeneous membrane-derived vesicles that are actively secreted by various forms of cell, and they can be visualised by electron microscopy [114]. Recent studies have reported the role of exosomes in immune regulation [115], intercellular communication [116,117] and biological events such as the coagulation [118] and microenvironmental regulation of tissues [119], as well as their role in the development of cancer, metastases and drug resistance [30,120].

In addition, a recent clinical trial found that the levels of exosomes were three to four times higher in the circulation of ovarian cancer patients compared to normal individuals [121]. As a result, a growing interest in determining the therapeutic importance of these nanoparticles in cancers has contributed to the discovery of either tissue or disease-specific exosome material, such as nucleic acids, proteins and lipids, as a source of new biomarkers [47]. While exosomes are known to be ideal biomarkers in the diagnosis of cancer due to their unique characteristics, there is still a long way to go in developing exosome-based assays.

#### 3.2.3. lncRNA and mRNA Biomarkers

Recent research has suggested that certain transcriptomes contain noncoding RNAs, such as long noncoding (lncRNAs) and microRNAs (miRNAs), as well as coding RNAs (mRNAs) as modern multiomics for the production of predictive, preventive and personalised medicine (PPPM) for ovarian cancers [48,122]. PPPM has been found to be an effective and affordable strategy for ovarian cancer care [123].

The first comprehensive study, conducted in 2019, investigated the lncRNA–miRNA–mRNA networks and lncRNA–RNA binding protein-mRNA networks in ovarian cancer, and confirmed the presence of some lncRNAs and mRNAs in ovarian cancer cell models [48].

These may be important sources for developing new biomarkers and anticancer targets for early stage detection and the successful treatment of ovarian cancer patients. However, further research and clinical practice is needed to improve the understanding and application of transcriptomes (lnRNAs, miRNAs, and mRNAs) in the management of ovarian cancer.

In EOC, some lncRNAs were found to be differentially expressed compared with benign and normal tissues, which demonstrated the up- or downregulation of 663 lncRNAs [124]. Based on a systematic analysis of the profiles of lncRNA and mRNA expression from the Cancer Genome Atlas (TCGA), a platinum resistance-specific lncRNA-mRNA network was discovered involving a total of 124 significant lncRNA-mRNA coexpression relationships that primarily regulate metabolic pathways, indicating the prognostic and therapeutic potential of lncRNAs in high-grade serous ovarian cancer [125].

#### 3.2.4. Aldehyde Dehydrogenase 1 (ALDH1)

A member of the aldehyde dehydrogenase protein family, ALDH1A1 expressed in a subpopulation of tumour-initiating cells and is therefore a potential candidate biomarker for cancer therapy. In a recent study by Chang et al. utilising an immunohistochemistry staining microarray study, the possible functions of ALDH1 in ovarian cancer consisted of the diagnosis of tumour type and disease staging, as well as therapeutic responses and overall survival rate. The data have shown that ALDH1 expression has been linked with longer average patient survival and that elevated ALDH1 expression is a positive prognosis factor in patients with ovarian cancer [126].

Comparably, a more recent analysis assessing the expression of ALDH1 in EOC stem cells found a higher expression of ALDH1 in CD44^+^ stem cell clones [49]. Hence, ALDH1 may be a useful biomarker for the identification of tumorigenic stem cells. Similarly, high ALDH1 expression in tumour cells was significantly associated with histological subtypes, the early FIGO stage, a well-differentiated grade and better survival probability (*p* < 0.05). The expression of ALDH1 in stromal cells had no clinicopathological associations in the present study (*p* > 0.05) [127].

#### 3.2.5. Folate Receptor Alpha (FOLR1)

FOLR1 is a membrane-bound receptor protein that is active in the movement of folate to cells and other cellular processes. The overexpression of FOLR1 was found in 69% of uterine serous carcinoma [128] and rapidly dividing cells. The expression of FOLR1 is regulated by the loss of extracellular folate levels, the accumulation of homocysteine, steroid hormone levels and genetic mutations. An initial study suggested a significant correlation between folate levels and tumour aetiology, and folate levels and progression, with suggestions for future research in FOLR1 gene expression and regulation [129].

In addition, the overexpression of FOLR1 has been documented in various epithelial nonmucinous tumours, including ovarian carcinoma; however, its assessment as a novel biomarker for early detection has not yet been verified. In either case, the overexpression of FOLR1 was reported in serous ovarian carcinoma describing clinicopathological characteristics and outcomes, as well as the relationship between FOLR1 and chemoresistance [50].

#### 3.2.6. Glutathione S-Transferase Polymorphisms

Members of the Glutathione S-transferase family (GSTM1, GSTT1, and GSTP1) are the result of major structural gene deletions, which in turn control the metabolism of drugs and impair chemotherapy in cancer patients. The GST polymorphisms are highly expressed in human ovaries [130]. Earlier epidemiologic studies did not confirm the association of GST polymorphisms with epithelial ovarian cancer [131], while it was proposed that in persons with homozygous deletions of GSTM, where GSTT had decreased or where there was no GST involvement, the removal of electrophilic carcinogens was challenging.

In addition, a recent analysis utilising DNA extracts from epithelial ovarian cancer tissue in which the GSTT1, GSTM1 and GSTP1 genotypes were identified using multiplex PCR and PCR-RFLP indicated that the combination of no GSTM1 and low GSTP1 resulted in over 60% improvement in progression-free survival and almost 40% improvement in overall survival [132]. Similarly, a meta-analysis investigating the relationship of GST polymorphisms with ovarian cancer risk suggested that the role of GSTs is highly significant in drug-resistant tumours where the higher expression of GSTs could alter the control of the kinase cascade during drug therapy [133].

## 4. Summary and Discussion

Despite both the conventional and new methods used to detect the development of ovarian cancer, such as radiographic imaging, invasive biopsies, tumour markers, and a combination of transvaginal ultrasound and tumour markers, ovarian cancer remains the most common gynaecological malignancy and has the highest mortality rate. The identification and confirmation of early warning biomarkers that are particularly specific to ovarian cancer involve the creation of minimally invasive screening approaches to identify the early onset of ovarian cancer. A summary of newly emerging ovarian cancer biomarkers highlighting their primary location in human body fluids is illustrated in Figure 2. The evaluation of promising early detection biomarkers opens up broad horizons for the detection and treatment of ovarian cancer.

Human serum proteome research has offered improved biomarker candidates for early detection, which is a crucial advancement as early diagnosis increases the five-year survival rate by more than 90%. CA125 is a well-studied high-discriminative tumour marker, especially among postmenopausal women, and it rises well before the emergence of clinical symptoms. However, it is recognised that the spike in CA125 levels in other types of cancer, benign ovarian disease, endometriosis, inflammatory conditions and ovulation, as well as its low early stage sensitivity, limit its ability as a single biomarker for ovarian cancer screening. As a consequence, a multibiomarker panel is recommended to strengthen the sensitivity and specificity of CA125, in which CA125 is used with HE4, mesothelin or a number of combinations (See Section 2.3. Highlight of the Most Common Biomarkers Combination for Ovarian Cancer Management) through which greater sensitivity and specificity has been achieved. For example, HE4 and mesothelin have been the most promising candidates to date.

Häusler et al. demonstrated higher expressions of *miR-21, miR-141, miR-200a, miR-200c, miR-203, miR-205, and miR-214*, as well as similarities in miRNA profiling in exosomal *microRNA* in ovarian cancer patients, suggesting that *miRNA* profiling could be a possible new technology for the early detection of ovarian cancer, biopsy profiling, and screening in asymptomatic populations [134]. On the other hand, studies have shown that ALDH1-positive ovarian cancer cells enhance tumorigenicity and chemoresistance; thus, the early detection of ovarian carcinomas along with other markers of gene mutations in *BRCA1/BRCA2*, Prostasin (PRSS8), GSTT1, FOLR1, KLK6, KLK7, and ALDH1 (Table 2) can also be predicted, requiring further investigation and clinical trials.

## 5. Future Directions

Numerous genomic and proteomic profiling experiments have provided valuable mechanistic knowledge on the development and evolution of new ovarian cancer biomarkers. However, several profiling experiments have been conducted on a small number of samples underdefined parameters, while some other studies display minimal overlap and are not necessarily reproducible. In order to verify the discriminatory influence of these emerging biomarkers, a large number of samples with multiple and separate sample sets would be needed. Promising biomarkers proposed in this study include the Cu isotope, exosomes, GSTT1, FOLR1, ALDH1, and mRNAs, along with multibiomarker panels in conjunction with CA125, as the most developed and efficient biomarkers currently available for clinical use.

## 6. Conclusions and Recommendations

Ovarian cancer is a high-mortality gynaecological condition affecting women all over the world. While substantial improvement has been made in the detection and overall five-year survival rate of ovarian cancer patients, both rates remain very low. This is because successful early stage diagnostic biomarkers and therapeutic goals remain inadequate. It is also important to identify novel early detection molecules or therapeutic targets, including pathways with a minimally invasive approach and a high sensitivity and specificity, which can greatly increase the overall survival rate and quality of life of ovarian cancer patients.

## Figures and Tables

**Figure 1 diagnostics-11-00465-f001:**
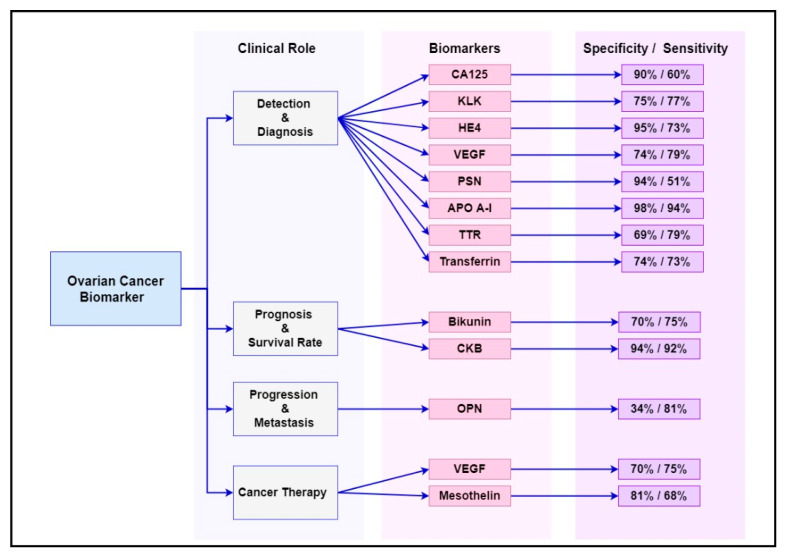
Summary of the sensitivity and specificity of the most common ovarian cancer biomarkers at early stages (stage I–II) of the disease.

**Figure 2 diagnostics-11-00465-f002:**
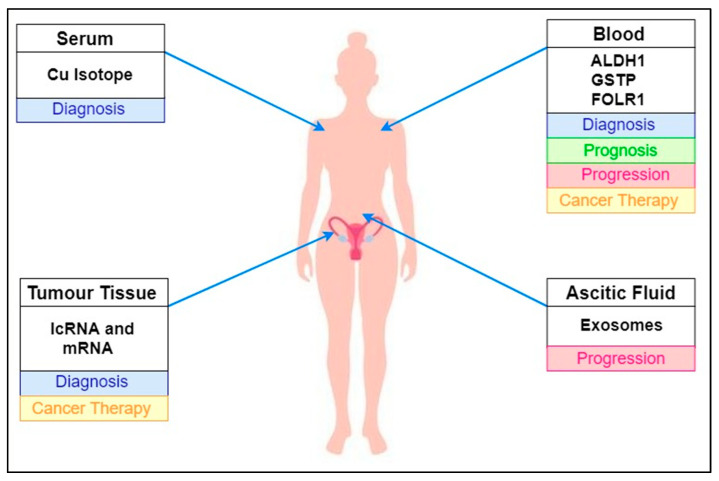
Summary of newly emerging ovarian cancer biomarkers highlighting their primary location in human body fluids.

**Table 1 diagnostics-11-00465-t001:** Current ovarian cancer biomarkers and their clinical utility.

Clinical Biomarker	Source	Clinical Role			Clinical Utility	Reference
		Diagnosis	Prognosis	Cancer Therapy		
Carbohydrate Antigen 125 (CA125)	Serum	✓	✓		Screen for ovarian cancer; evaluate the chemotherapy response and monitor disease recurrence using ELISA.	[12,13,14,15]
Osteopontin (OPN)	Plasma	✓	✓		Diagnosis of ovarian cancer and prognostic indicator of metastasis using PCR and ELISA.	[16,17]
Kallikreins (KLKs)	Serum	✓			Diagnosis of ovarian cancer using ELISA.	[17,18,19,20,21]
Bikunin	Serum		✓		Predict the stage of the disease and the survival rate preoperatively using ELISA or immunoblot assay.	[22,23,24,25]
Human Epididymis Protein 4 (HE4)	Serum	✓			High-sensitivity diagnostic tool for detecting stage I ovarian cancer using ELISA.	[26,27,28]
Vascular Endothelial Growth Factor (VEGF)	Serum	✓	✓	✓	Indicator of a shorter survival time in patients with ovarian cancers using ELISA.	[29,30,31,32]
Prostasin (PSN)	Serum	✓			Identify patients with ovarian cancer through RT-PCR.	[33,34]
Creatine Kinase B (CKB)	Serum		✓		Predict survival rate and prognosis of ovarian cancer using microarray technology.	[35]
Mesothelin	Serum			✓	Detected using ELISA method and used clinically as an antimesothelin immunotoxin therapy to attack cancer cells in combination with standard chemotherapy.	[36,37,38]
Apolipoprotein A-I (apoA-I)	Plasma	✓			Detected clinically using ELISA method to confirm diagnosis with ovarian cancer.	[39,40,41,42]
Transthyretin (TTR)	Serum	✓			Diagnosis of early stage ovarian cancer patients using matrix-assisted laser desorption-ionisation (MALDI).	[43]
Transferrin	Serum	✓			Used in combination with CA125 as an improved and sensitive ovarian cancer detection method estimated by measurements of the serum total iron-binding capacity (TIBC).	[44,45]

**Table 2 diagnostics-11-00465-t002:** Potential newly emerging biomarkers and their clinical utility in ovarian cancer care.

Emerging Biomarkers	Source	Potential Clinical Role	Reference
Cu isotope	Serum	Early diagnostic tool	[46]
Exosomes	Ascites	Cancer progression	[47]
lncRNA and mRNA	Tumour tissue	Early diagnostic tool and cancer therapy	[48]
Aldehyde Dehydrogenase 1 (ALDH1)	Blood/Cytosol	Early diagnostic and cancer progression	[49]
Folate Receptor Alpha (FOLR1)	Blood/Gene	Progression and cancer therapy	[50]
Glutathione S-Transferase Polymorphisms (GSTP)	Blood/DNA	Anticancer drug response	[51]

## Data Availability

No new data were created or analysed in this study. Data sharing is not applicable to this article.

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
