# Peer review of "New Predictive Biomarkers for Ovarian Cancer"

_diagnostics, 2021, doi:10.3390/diagnostics11030465_

Round 1

Reviewer 1 Report

In this review, the authors have introduced existing biomarkers and also suggested potential biomarkers in the future related to diagnostic, prognostic, therapeutic perspectives in ovarian cancer. The vast amount of searched information has been organized well. However, the required information is missing from the flow of the content. It is recommended the supplementation of the contents taking into account the following issues.

1. The author provided the narrative at the simple level of listing the theoretical information for each biomarker. Since the authors pointed out the shortcomings of the clinical use of the existing biomarkers comprehensively, it is necessary to provide corresponding information for each marker as to what disadvantages exist.
2. No information on real-world data in clinical was provided anywhere. In each narrative paragraph of existing biomarkers, the current state of practical use in clinical laboratory tests of diagnosis or prognosis including success and failure rate by each biomarker should be provided.
3. The section for the introduction of the new biomarker is also in the same context as pointed out in Comment 2. Further, the new suggested markers are that already being considered for use, unlike the title described by the author. However, there are obvious limitations in applying these new biomarkers to clinical actually. these points are missing in related paragraphs.

Reviewer 2 Report

The review is well written, structured, and illustrated. However, there are several important remarks in my opinion. Thus, not all existing markers of ovarian cancer are mentioned. The significance of such markers as prostazin, creatine kinase B, mesothelin, apolipoprotein A-I, transthyretin and transferrin, and others is known. The possibility of combining different markers has not been described, which significantly increases the diagnostic sensitivity and specificity. For example, when determining osteopontin, the sensitivity was 81%, when determining CA-125, the sensitivity was 84%, and when these two markers were combined, the sensitivity increased to 94%. The combined use of HE-4 and CA-125 also significantly increases the detection sensitivity of ovarian cancer. For what stage of ovarian cancer are the sensitivity and specificity values ​​given in Figure 1? I think it would be appropriate to give the appropriate values ​​for both the early and advanced stages. For new markers, the values ​​of sensitivity and specificity are not shown in Figure 2 (by analogy with Figure 1). And most importantly, there is no discussion, only a listing of markers. A small section should be added to summarize the information.

Round 2

Reviewer 1 Report

All concerns have been well addressed. No additional comment to raise.

Reviewer 2 Report

The authors took into account all the comments of the reviewers and significantly revised the article. I believe that it is possible to recommend the article for publication in its current form.